# EMBEDDING-BASED STATISTICAL INFERENCE ON GENERATIVE MODELS

## ABSTRACT

Generative models are capable of producing human-expert level content across a variety of topics and domains. As the impact of generative models grows, it is necessary to develop statistical methods to understand the population of available models. These methods are particularly important in settings where the user may not have access to information related to a model's pre-training data, weights, or other relevant model-level covariates. In this paper we extend recent results on representations of black-box generative models to model-level statistical inference tasks. We demonstrate – both theoretically and empirically – that the use of these representations are effective for multiple model-level inference tasks.

## 1 INTRODUCTION

Generative models have recently met or surpassed human-level standards on benchmarks across a range of tasks (Nori et al., 2023; Katz et al., 2024; Dubey et al., 2024). While these claims warrant skepticism and further robustness evaluation (Ness et al., 2024), the impressive capabilities have created a competitive environment for training state-of-the-art models and have inspired the development of complementary methods to adapt models to particular use cases. For example, quantizing the weights of a neural model enables a trade-off of model precision with vRAM and disk space (Gholami et al., 2022) while methods such as Low Rank Adaptation (LoRA) (Hu et al., 2021) and prompt-tuning (Lester et al., 2021) enable compute- and data-efficient model adaptation. Other methods such as retrieval-augmented generation (Lewis et al., 2020), model merging (Matena & Raffel, 2022), constrained decoding (Hokamp & Liu, 2017), etc. (Dettmers et al., 2024; Edge et al., 2024), have similarly contributed to the rapid development of a large population of diverse and accessible models.

Each model in the population has an accompanying set of covariates – scores on benchmarks, training mixture proportions, model safety scores, etc. – that are a function of the model, the training set, the architecture, the retrieval database, or derivatives thereof. For a given model, the covariate of interest may not be available to the user or it may be too expensive to calculate. For example, it may not be known if a proprietary model has been trained on copyrighted data. Or, it may be too resource intensive to directly evaluate how toxic every model is. Methods for predicting model-level covariates are necessary in these settings, and others, to fully understand the behavior and properties of a model.

In this paper we extend recent theoretical results for vector representations of generative models (Acharyya et al., 2024) to model-level statistical inference settings. Our results show that the embeddings of the responses from a collection of generative models can be used for consistent inference for a wide class of inference problems. We demonstrate the effectiveness of the representations for three downstream inference tasks, including predicting the presence of sensitive information in a model's training mixture and predicting a model's safety. We include empirical investigations of performance sensitivity to hyperparameters required to generate the model-level vector representations.

**Contribution.** Our contribution is the theoretical and empirical validation for using vector representations of black-box generative models for model-level inference tasks. The representations that we study herein are based on the embeddings of their responses. The theoretical results apply to any generic, well-behaved embedding function. Given the representations, any standard vector-based method can be used for inference on the models.

## 1.1 Background & related work

Our work is an extension of recent theoretical and empirical results on embedding-based representations of generative models (Acharyya et al., 2024), which itself is a continuation of a long-line of embedding-based investigations of the inputs and outputs of generative models (Mikolov, 2013; Reimers, 2019; Neelakantan et al., 2022; Patil et al., 2023). Of particular relevance is recent empirical work defining the data kernel (Duderstadt et al., 2023) and investigations into its ability to track the dynamics of interacting models (Helm et al., 2024), in which the experiments demonstrate the ability to parlay the embeddings of a collection of inputs or outputs into useful vector representations of the generative models themselves in both white-box and black-box settings. The results herein further theoretically and empirically validate these findings in the context of statistical inference.

Our work is also related to using embedding-based techniques for inference on complicated objects such as entire mouse connectomes (Wang et al., 2020), physiological data (Chen et al., 2022), and classification distributions (Helm et al., 2021). In each of these settings, the authors define a pairwise distance matrix on the objects and apply multi-dimensionsal scaling to obtain vector representations of each of the objects. Once there is one vector representation per object, standard inference methods for the specific task can be used. The method we study herein follows this general formula, with the additional complication that the objects are random mappings.

Lastly, our work is a part of the relatively new literature on inference on generative models. For example, FlashHELM (Perlitz et al., 2023) uses the score on a subset of a benchmark such as HELM (Liang et al., 2022) to quickly identify where a model fits on a leaderboard. More recent work proposes scaling laws to predict the performance of base models on a suite of benchmarks as a function of training FLOPs (Ruan et al., 2024). The method proposed herein does not assume access to a scoring function nor does it assume access to a function of the model's weights at inference time. Perhaps most importantly, our setting and method can be applied to general model-level prediction.

## 2 The Data Kernel Perspective Space

Before we present our results related to statistical inference on black-box generative models, we first describe how to obtain finite-dimensional vector representations of the models. The particular method for obtaining vector representations for generative models that we study herein has previously been discussed in Helm et al. (2024) and Acharyya et al. (2024).

Let $f : \mathcal{Q} \to \mathcal{X}$ be a black-box model from a query space $\mathcal{Q}$ to an output space $\mathcal{X}$. In our setting there are $n$ models $f_1, \ldots f_n$ and $m$ queries $q_1, \ldots, q_m$ with $q_j \in \mathcal{Q}$. We assume that each model responds to every query $r$ independent times and let $f_i(q_j)_k$ be the $k$-th response to $q_j$ from $f_i$.

We let $g : \mathcal{X} \to \mathbb{R}^p$ be an embedding function that maps a model response to a $p$-dimensional real-valued vector. Further, we let $x_{ijk} = g(f_i(q_j)_k)$ denote the embedding of $f_i(q_j)_k$ and $F_{ij}$ denote the distribution of $x_{ijk}$ on $\mathbb{R}^p$. That is, $x_{ij1}, \ldots x_{ijr} \sim^{iid} F_{ij}$.

We let $\bar{x}_{ij} = \frac{1}{r} \sum_{k=1}^{r} x_{ijk}$ be the empirical average embedded response of $f_i$ to $q_j$ and $\bar{X}_i \in \mathbb{R}^{m \times p}$ denote the matrix whose $j$-th row is $\bar{x}_{ij}$. We view $\bar{X}_i$ as a matrix representation of model $f_i$ with respect to $\{q_1, \ldots, q_m\}$ and $g$. We define $D$ as the $n \times n$ pairwise distance matrix with entries

$$D_{ii'} = \frac{1}{m} \big|\big| \bar{X}_i - \bar{X}_{i'} \big|\big|_F. \tag{1}$$

The matrix entry $D_{ii'}$ captures the difference in empirical average model behavior between $f_i$ and $f_{i'}$ with respect to $\{q_1, \ldots, q_m\}$ and $g$.

With the form described by Eq. (1), $D$ is a rescaled Euclidean distance matrix. Hence, multi-dimensional scaling (MDS) of $D$ yields $d$-dimensional Euclidean representations of the matrices $\bar{X}_i$ (and thereby of the models $f_i$) with respect to $\{q_1, \ldots, q_m\}$ and $g$ (Torgerson, 1952). Letting $\widehat{\psi} := \text{MDS}(D) \in \mathbb{R}^{n \times d}$, we refer to $\widehat{\psi}$ as the *data kernel perspective space* (DKPS). We emphasize that the $i$-th row of the DKPS $- \widehat{\psi}_i -$ is a $d$-dimensional vector representation of the generative model $f_i$.

## 2.1 ANALYTICAL PROPERTIES OF THE DATA KERNEL PERSPECTIVE SPACE

While the DKPS representations of the models are Euclidean objects, it is not possible to comment on their properties without imposing constraints on the $F_{ij}$. We let $\mu_i \in \mathbb{R}^{m \times p}$ be the population counterpart of $\bar{X}_i$ whose $j$-th row is $\mathbb{E}_{x \sim F_{ij}}(x)$. Similarly, we let $\Delta$ be the population counterpart of $D$ whose $(i, i')$-th element is $\Delta_{ii'} = \frac{1}{m}||\mu_i - \mu_{i'}||_F$. We assume $\Delta^*_{ii'} = \lim \frac{1}{m}||\mu_i - \mu_{i'}||_F$ for every pair $(i, i')$ as $m, r \to \infty$ and that $g$ is bounded.[1]

In our setting, under appropriate assumptions described in (Acharyya et al., 2024),

$$D_{ii'} = \frac{1}{m} \left\| \bar{X}_i - \bar{X}_{i'} \right\| \to \Delta^*_{ii'} \tag{2}$$

with high probability as $m, r \to \infty$ for all $(i, i') \in \{1, \ldots, n\} \times \{1, \ldots, n\}$. Further, there exists $\psi := \text{MDS}(\Delta^*) \in \mathbb{R}^{m \times d}$ such that $\widehat{\psi} \to \psi$. That is, the configuration $\psi$ is the population counterpart of $\widehat{\psi}$. $\psi$ captures the true geometry of the mean discrepancies of the model responses with respect to the queries $\{q_1, \ldots, q_m\}$. Importantly, in settings where the models are not fixed and are instead assumed to be *i.i.d.* realizations from a model distribution $P_{model}$, the distance matrix $D$ converges to $\Delta^*$ and, under technical assumptions, there exists $\psi$ such that $\widehat{\psi} \to \psi$ as $m, r \to \infty$ for all $n$ (Acharyya et al., 2024).

## 2.2 STATISTICAL INFERENCE IN THE DATA KERNEL PERSPECTIVE SPACE

We now extend the consistency of $\widehat{\psi}$ to $\psi$ to model-level inference. Consider the classical statistical learning problem (Hastie et al., 2009, Chapter 2) in the context of a collection of generative models: given training data $\mathcal{T}_n = \{(f_1, y_1), \ldots, (f_n, y_n)\}$ assumed to be *i.i.d.* realizations from the joint distribution $P_{fY}$, choose the decision function $h : \mathcal{F} \to \mathcal{Y}$ that minimizes the expected value of a loss function $\ell$ with respect to $P_{fY}$ within a class of decision functions $\mathcal{H}$ for a test observation $f$ assumed to be an independent realization from the marginal distribution $P_f$. Or, with $\mathcal{R}_\ell(P_{fY}, h) := \mathbb{E}_{P_{fY}} [\ell(h(f), y)]$, select $h^*$ such that

$$h^* \in \underset{h \in \mathcal{H}}{\text{argmin}} \, \mathcal{R}_\ell(P_{fY}, h).$$

We let $\mathcal{R}^*_\ell(P_{fY}, \mathcal{H}) = \mathcal{R}_\ell(P_{fY}, h^*)$ denote the expected loss (or "risk") of $h^*$.

The joint distribution $P_{fY}$ is often unavailable and the decision function is selected based on the training data. We let $h(\,\cdot\,; \mathcal{T}_n)$ denote such a decision function and say *the sequence of decision functions* $(h(\,\cdot\,; \mathcal{T}_1), \ldots, h(\,\cdot\,; \mathcal{T}_n))$ *is consistent for* $P_{fY}$ *with respect to* $\mathcal{H}$ *if*

$$Pr\big(\, \big|\mathcal{R}_\ell(P_{fY}, h(\,\cdot\,; \mathcal{T}_n)) - \mathcal{R}^*_\ell(P_{fY}, \mathcal{H})\big| > \epsilon\big) \to 0$$

as $n \to \infty$.

In the black-box setting we do not have direct access to useful representations of the models. Instead, we can use the representations $\{\psi_1, \ldots, \psi_n\}$ as proxies for the models $\{f_1, \ldots, f_n\}$. Hence, $\mathcal{T}_n = \{(\psi_1, y_1), \ldots, (\psi_n, y_n)\}$ where $(\psi_i, y_i) \sim^{iid} P_{\psi Y}$ and our goal is to choose a decision function $h^*$ which minimizes the average loss $\mathcal{R}_\ell(P_{\psi Y}, h) := \mathbb{E}_{P_{\psi Y}} [\ell(h(\psi), y)]$ within an appropriately defined $\mathcal{H}$. Note that the true $\{\psi_1, \ldots, \psi_n\}$ are not known *a priori* and must be estimated via $\{\widehat{\psi}_1, \ldots, \widehat{\psi}_n\}$. We let $\widehat{\mathcal{T}}_n$ be the training data where $\psi_i$ is replaced with $\widehat{\psi}_i$.

Two important extensions of the DKPS consistency results in the context of the classical statistical learning problem are the following theorems:

**Theorem 1.** *Under technical assumptions described in Appendix A,*

$$\mathcal{R}_\ell(P_{\psi Y}, h(\,\cdot\,; \widehat{\mathcal{T}}_n)) \to \mathcal{R}_\ell(P_{\psi Y}, h(\,\cdot\,; \mathcal{T}_n))$$

*as $m, r \to \infty$, for every $n$.*

---

[1] Boundedness is typically satisfied in practice for well-defined $g$. For language models, an element of $\mathcal{X}$ is a finite sequence from a finite vocabulary; for text-to-image models $\mathcal{X} = \{0, \ldots, 255\}^3$ is finite.

That is, the risk of a decision function based on $\widehat{\mathcal{T}}_n = \{(\widehat{\psi}_1, y_1), \ldots, (\widehat{\psi}_n, y_n)\}$ converges to the risk of the decision function based on the true-but-unknown $\mathcal{T}_n$. For situations where $\{\psi_1, \ldots, \psi_n\}$ are good proxies, Theorem 1 states that increasing the number of queries and/or number of replicates per query will improve the performance of decision functions trained on $\widehat{\mathcal{T}}_n$.

**Theorem 2.** *Under technical assumptions described in Appendix A, if $(h(\,\cdot\,; \mathcal{T}_1), \ldots, h(\,\cdot\,; \mathcal{T}_n))$ is consistent for $P_{\psi Y}$ with respect to $\mathcal{H}$, then $(h(\,\cdot\,; \widehat{\mathcal{T}}_1), \ldots, h(\,\cdot\,; \widehat{\mathcal{T}}_n))$ is consistent for $P_{\psi Y}$ with respect to $\mathcal{H}$ as $n, m, r \to \infty$.*

Theorem 2 states that if the sequence of decision functions learned from $\mathcal{T}_n$ is consistent then the sequence of decision functions learned from the analogous $\widehat{\mathcal{T}}_n$ is also consistent. This result suggests that inference performance should improve with more training data. While a logical follow-on to Theorem 1 and a well-understood principle in classical statistical learning settings, the consistency of the DKPS in the setting where the number of models grows is a non-trivial extension of the fixed $n$ case.

The proofs of Theorems 1 and 2 are provided in Appendix A.

### 2.2.1 NON-STANDARD CONSIDERATIONS

Given that our analysis is based on estimates $\{\widehat{\psi}_1, \ldots, \widehat{\psi}_n\}$ of proxies $\{\psi_1, \ldots, \psi_n\}$ of the objects of interest $\{f_1, \ldots, f_n\}$, it is important to note some non-standard theoretical considerations. To facilitate our discussion we let $\mathcal{R}_\ell^*(P_{fY}) := \inf_{\{h : \mathcal{F} \to \mathcal{Y}\}} \mathcal{R}_\ell(P_{fY}, h)$ be the Bayes optimal risk of $P_{fY}$. Further, we let $P_{query}$ be a distribution on queries.

For example, while our theoretical results provide insights as to how performance will be affected by increasing $n, m,$ and $r$, our results do not comment on the magnitude of $|\mathcal{R}_\ell^*(P_{\psi Y}) - \mathcal{R}_\ell^*(P_{fY})|$. This quantity is a measure of how good of a proxy $\{\psi_1, \ldots, \psi_n\}$ is for $\{f_1, \ldots, f_n\}$. In particular, if $|\mathcal{R}_\ell^*(P_{\psi Y}) - \mathcal{R}_\ell^*(P_{fY})| = 0$ then the $\psi_i$ are perfect proxies of the $f_i$ with respect to $y$ (Devroye et al., 2013, Chapter 32). We expect it to be challenging to theoretically understand $|\mathcal{R}_\ell^*(P_{\psi Y}) - \mathcal{R}_\ell^*(P_{fY})|$ in general.

Similarly, let $q_1, \ldots, q_m \sim^{iid} P_{query}$ and $q_1', \ldots, q_m' \sim^{iid} P_{query}'$ with $P_{query} \neq P_{query}'$. The proxies of the models induced by these two query sets – $\{\psi_1, \ldots, \psi_n\}$ and $\{\psi_1', \ldots, \psi_n'\}$, respectively – will be of different quality. Without an initial understanding of the models and their relationship with the covariates, we expect that it will be impossible to know which query distribution is preferred *a priori*. We highlight the importance of $P_{query}$ in an experimental setting below.

Lastly, we note that representations of the models that we analyze herein capture the intrinsic geometry of the mean discrepancies of the models for the queries $\{q_1, \ldots, q_n\}$. For covariates that cannot be described as a function of the mean discrepancies of the models, it is unclear how to interpret Theorems 1 and 2. We discuss potential alternatives to the representations that we study in the discussion.

### 2.3 AN ILLUSTRATIVE EXAMPLE – "WAS RA FISHER GREAT?"

The first model-level inference task we study is a toy example where the task is to predict the probability that a model will respond "yes" to the question "Was RA Fisher great?". The question is chosen due to its subjectiveness – there is no correct answer to the question of a person's greatness – and its duality – Ronald A. Fisher is considered one of the most influential statisticians in history and is considered an advocate of the 20th century Eugenics movement (Bodmer et al., 2021).

We use a 4-bit version of Meta's `LLaMA-2-7B-Chat` (Touvron et al., 2023) as a base model and consider a collection of models parameterized by fixed context augmentations. Each augmentation $a_i$ contains information related to RA Fisher's statistical achievements (i.e., $a_i$ = "RA Fisher pioneered the principles of the design of experiments") or to his involvement in the eugenics movement (i.e., $a_i$ = "RA Fisher's view on eugenics were primarily based on anecdotes and prejudice.") and is prepended to every query. The covariate corresponding to a given model is calculated by prompting the base model with the appropriately formatted prompt "Give a precise answer to the question based on the context. Don't be verbose. The answer should be either a yes or a no. $a_i$. Was RA Fisher great?" until there are 100 valid responses. We let $y_i$ be the average number of "yes"es.

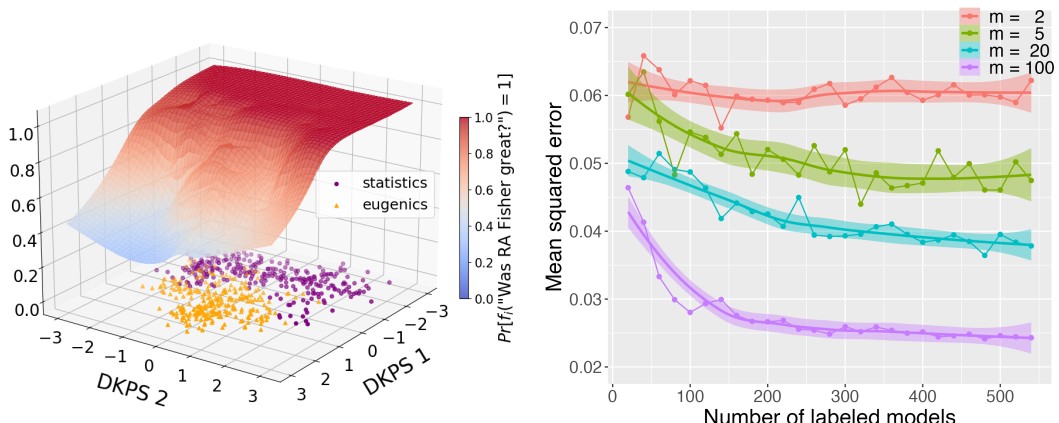

Figure 1: **Left.** The 2-d Data Kernel Perspective Space (DKPS) and covariate surface for a collection of 550 models parameterized by fixed augmentations. **Right.** The performance of the 1-nearest neighbor regressor in DKPS for predicting the probability that an unlabeled model responds "yes" to "Was RA Fisher great?".

To induce a DKPS for this task, we consider queries sampled from OpenAI's ChatGPT with the prompt "Provide 100 questions related to RA Fisher.". For a given query $q_j$ we prompt the base model with the appropriately formatted prompt "$a_i$ $q_j$" and fix $r = 1$. The left figure of Figure 1 is a 3-d figure where the first two dimensions are the DKPS of the $n = 550$ (275 statistics augmentations, 275 eugenics augmentations) models induced with $m = 100$ queries. Model responses are embedded by averaging the per-token last layer activation of the base model. The third dimension is an interpolated $y_i$ surface with a linear kernel (Du Toit, 2008). The first dimension of the DKPS is clearly capable of distinguishing between models adorned with "statistics" augmentations and models adorned with "eugenics" augmentations. The shape of the interpolated covariate surface is highly correlated with this feature. A description of the augmentations that parameterize the models and the queries used to induce the DKPS is provided in Appendix B.1.

The right figure of Figure 1 shows the performance of a 1-nearest neighbor (1-NN) regressor in DKPS for a varying number of labeled models and a varying number of queries. The DKPS is induced with $n$ models and the regressor is trained with $n - 1$ of the model-level covariates. The mean squared error reported is the error of the 1-NN regressor for predicting the "left out" model's covariate ($\pm$ 1 S.E.). As Theorems 1 and 2 suggest, the performance of the regressor is dependent on both the number of models and the number of queries: the more models and the more queries the better. The scale of the impact of more models and more queries, however, depends on its counterpart. The amount of models does not have a large impact on predictive performance if the number of queries is small. The amount of queries has a large impact on performance regardless of the number of models.

## 3 EXPERIMENTS

We next consider two experiments with more realistic model-level covariates: predicting whether or not a model has had access to sensitive information and predicting model safety. For all experiments we fix $r = 1$ as suggested by the empirical rates of convergence of the perspectives described in Acharyya et al. (2024). We use the MDS implementation from Graspologic (Chung et al., 2019) throughout. We use the profile likelihood of the singular values of $D$ to determine the dimensionality of the DKPS (Zhu & Ghodsi, 2006) and note that this may be larger than two. We show only the first two dimensions for visualization purposes.

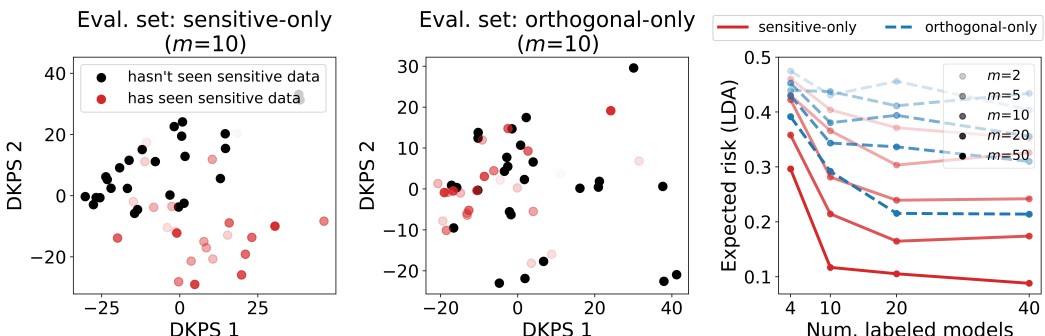

Figure 2: **Left.** The 2-d data kernel perspective space (DKPS) of 50 fine-tuned models – 25 with "sensitive" data in the fine-tuning data mixture (red), 25 with none (black) – induced by an evaluation set containing 10 prompts relevant to the sensitive data. For models trained on sensitive data, color intensity correlates with amount of sensitive data in the training mixture. **Center.** The 2-d DKPS of the models induced by a set of 10 prompts "orthogonal" to the difference between models with sensitive data in their fine-tuning data mixture and models with no sensitive data in their fine-tuning data mixture. **Right.** Classification performance as a function of number of labeled models and size of evaluation set for both sensitive and orthogonal evaluation sets.

### 3.1 HAS A MODEL SEEN SENSITIVE INFORMATION?

Modern language models are trained with trillions of tokens of text (Touvron et al., 2023). For proprietary models such as OpenAI's GPT series or Anthropic's Claude series, the exact sources of the training mixtures are unknown and – given the models' propensities to produce content that is strikingly similar to copyrighted content (Henderson et al., 2023) – its curation and use is ethically questionable (Lemley & Casey, 2020). Further, the training mixture of some models may include sensitive informative such as personal information, trade secrets, or government-classified information that should never be presented to the end user. Developing classifiers to identify models that are either trained on sensitive or copyrighted information or are likely to produce sensitive or copyrighted information is thus paramount to uphold the rights of the stakeholders of the original content. There has been recent work on this topic in settings with access to model weights or token likelihoods (see, e.g., (Duderstadt et al., 2023; Shi et al., 2023)).

To investigate the utility of DKPS for this purpose, we again use a 4-bit version of `LLaMA-2-7B-Chat` as a base model and train 50 different LoRA adapters with different subsets of the Yahoo! Answers (YA) dataset (Zhang et al., 2015). The YA dataset consists of data from 10 topics. We consider data from the topic "Politics & Government" to be "sensitive" information, data from the topics "Society & Culture", "Science & Mathematics", "Health", "Education & Reference", "Computers & Internet", and "Sports" to be "not-sensitive", and data from the remaining topics to be "orthogonal". We trained each of the 50 adapters with 500 question-answer pairs for 3 epochs with a learning rate of $5 \times 10^{-5}$ and a batch size of 8. Each adapter is rank 8, has a scaling factor of 32, targets all attention layers, does not have bias terms, and has a dropout probability of 0.05 when training. For 25 of the adapters, the adapter training mixture consisted wholly of randomly selected not-sensitive data. For the remaining 25 adapters, the adapter training mixture consisted of $p_i$ randomly selected sensitive data and $500 - p_i$ randomly selected not-sensitive data. We let $y_i$ be the indicator of whether or not the adapter's training mixture contained any sensitive data.

We study classification of the models in two different DKPS: one induced by a set of randomly selected sensitive queries, one induced by a set of randomly selected orthogonal queries. Both DKPS use the open source embedding model `nomic-embed-v1.5` (Nussbaum et al., 2024). A 2-d DKPS induced by $m = 10$ randomly selected sensitive queries is shown on the left of Figure 2. In this space the models are separated by their label and the models that have seen more sensitive information are generally farther from the class-boundary. A 2-d DKPS induced by $m = 10$ orthogonal queries is shown in the center of Figure 2. Here, by contrast, the models are not easily separable by their label.

The right figure of Figure 2 shows the classification performance of Fisher's Linear Discriminant trained on varying amounts of DKPS representations of models for the two query distributions. For a given $n$ and $m$, we induce the DKPS with all models and a randomly selected query set. We report the expected risk of the classifier trained on a random subset of the models for the remaining models. We observe similar phenomena to the "Was RA Fisher great?" experiment in that both the amount of models and the amount of queries impact performance. For a fixed $m$, the expected risk curves also highlight the observed difference in separability of the models in the two DKPS, with the expected risk with sensitive queries being significantly lower than the expected risk with orthogonal queries. Indeed, the expected risk with $m = 10$ sensitive queries is similar to the expected risk with $m = 50$ orthogonal queries.

Lastly, we highlight the 1-d projection learned by Fisher's linear discriminant for $m = 10$ for both DKPS in Figure 3. As can be seen in the top figure of Figure 3, the projection of the models is correlated with the proportion of sensitive data that the model has had access to when the DKPS is induced with sensitive queries. While the linear goodness-of-fit is not large ($R^2 = 0.37$), the correlation is statistically significant per the hypothesis test using Kendall's rank correlation coefficient ($\tau = 0.42, p < 0.01$). The projection of the models when the DKPS is induced with orthogonal queries has a line of best fit with a negligible slope and a much smaller Kendall's rank correlation coefficient ($\tau = 0.08$). The second of which results in a $p$ value of $0.57$ – meaning we fail to reject the null that the amount of sensitive information the model has had access to is correlated with the learned 1-d projection.

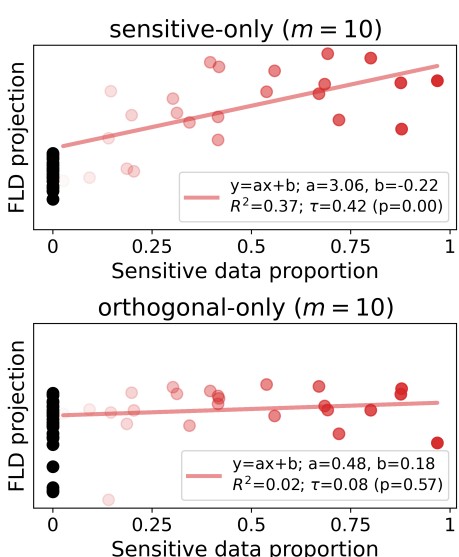

Throughout this experiment we use the term "orthogonal" for queries from topics that are not used when fine-tuning the models under study. The term is chosen because, naïvely, queries from these topics should not elicit different responses from models trained on sensitive data and models trained on not-sensitive data. In reality, the sensitive data and the not-sensitive data have different underlying token distributions and this difference will cause systematic differences in model responses after fine-tuning even for topics that are irrelevant *a priori*. Further, it is likely that the documents in the "orthogonal" topics share some content commonalities with documents in the sensitive and not-sensitive topics. We see this

Figure 3: **Top.** The 1-d FLD projection of the models from a DKPS induced by queries from the sensitive topic versus the amount of sensitive data the adapter had access to during training. **Bottom.** The same but for a DKPS induced by queries from orthogonal topics.

phenomenon in the classification results in the right figure of Figure 2 where the linear classifier is able to perform better than chance with enough "orthogonal" queries.

## 3.2 HOW SAFE IS A MODEL?

Model safety is one of the biggest concerns when deploying a language model in production. An unsafe model is prone to propagating harmful stereotypes (Ferrara, 2024), using toxic language (Wen et al., 2023), and misunderstanding the user's intent (Ji et al., 2023) – all of which can adversely affect the user and their experience. Hence, developing techniques to understand how unsafe a model is an important aspect of the model-production pipeline. As with predicting if a model has had access to sensitive information above, we investigate using DKPS to predict model safety through the lens of model toxicity and model bias.

We consider a collection of 58 models. Each model in the collection is a base model, a fine-tuned version of a base model, or the result of weight-merging various other models in the collection. We

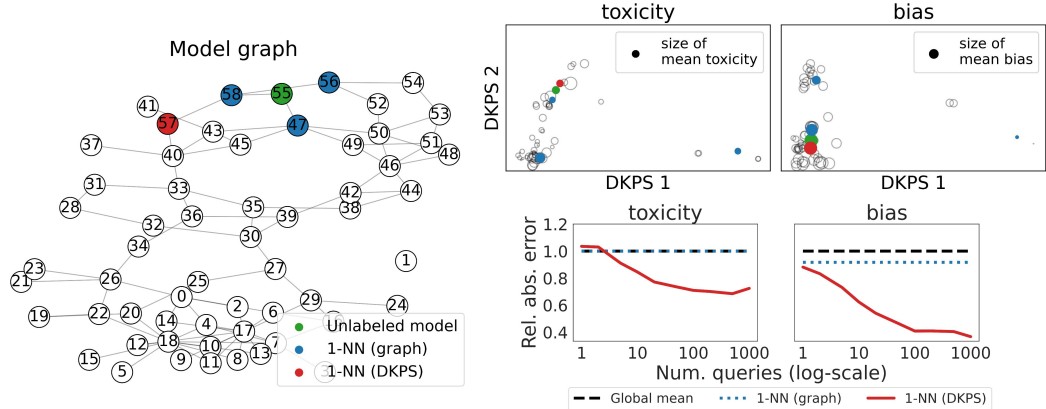

Figure 4: **Left.** A graph where each node is a model and an edge between two models exists if model $i$ is fine-tuned from model $i'$ or if model $i'$s weights were used in a model-merge that resulted in model $i$, etc. **Right, top.** The two-dimensional data kernel perspective spaces (DKPS) corresponding to the toxicity and bias prediction tasks. Dot size is proportional to model toxicity or bias. **Right, bottom.** Average relative performance of three regression techniques across all models in the model graph. Local predictions in DKPS are more effective than both global predictions and local predictions in model relationship space.

view this collection of models as a graph where each model is a node and an undirected edge exists between nodes if one of the models is a fine-tuned version of the other or if one of the models is the result of a model merge including the other. The graph representing the collection of models is shown on the left of Figure 4. The list of models under study is provided in Appendix B.2.

For each model in the collection we consider two covariates: model toxicity and model bias. To determine a model's toxicity, we prompt each model with a collection of queries from the Real Toxicity Prompts (RTP) (Gehman et al., 2020) dataset and subsequently evaluate the toxicity of each response with the neural model `roberta-hate-speech-dynabench-r4` (Vidgen et al., 2021). The model-level toxicity is simply the average response toxicity. An analagous process is used to define a model's bias with the dataset Bias in Open-ended Language Generation Dataset (BOLD) (Dhamala et al., 2021) and the `regard` model (Sheng et al., 2019).

We induce the perspective spaces for the toxicity and bias tasks with randomly sampled prompts from RTP and BOLD, respectively, and the embedding model `nomic-embed-v1.5`. The 2-d DKPS – induced with $m = 2000$ queries – for both tasks is shown in the top right of Figure 4. The size of the dot is correlated with the model's covariate. We highlight an example unlabeled model (green) and its corresponding neighbor in DKPS (red) and neighbors in graph-space (blue) in Figure 4. Importantly, the relative position of a model in the respective DKPS is predictive of the model's toxicity and the model's bias.

We quantify this observation by evaluating regressors for predicting the model-level toxicity and bias of an unlabeled model. We consider three different regressors for these tasks. The first is a constant equal to the average covariate of the labeled models (i.e., the "global mean"). The second uses the average covariate of models who share an edge with the unlabaled model

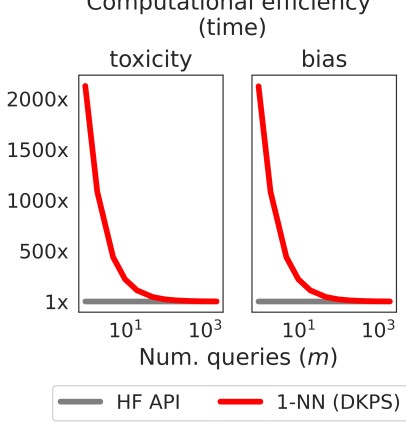

Figure 5: The relative time improvement (larger is better) when using local predictions in DKPS instead of calculating the ground-truth model-level covariate using HuggingFace's API.

(i.e., "1-NN (graph)"). The third uses the covariate of the nearest neighbor in DKPS (i.e., "1-NN (DKPS)"). For 1-NN (DKPS) we consider varying amounts of randomly sampled queries from RTP and BOLD to induce the DKPS. We use the global mean as a standard and report the relative absolute error of the three methods in the bottom right of Figure 4. The reported performance of the 1-NN (DKPS) regressor is the average of the smaller of $200$ and $2000/m$ random samples of $m$ queries. The reported relative absolute error is the average across all models in the model graph. Notably, given enough queries local predictions in DKPS outperform local predictions in the graph space and predictions using the global mean.

In addition to reporting the relative performance, we report the time it takes to use the DKPS machinery to predict `Mistralv1.0`'s (Jiang et al., 2023) toxicity and bias relative to using the evaluation model through HuggingFace's API (Wolf et al., 2019). The time we report for the HuggingFace API is the time it takes to calculate the "ground truth" used to calculate the performance of the regressors above and is the total time required for `Mistralv1.0` to produce responses and for the evaluation model to produce scores for 2000 queries. The time we report for 1-NN (DKPS) includes the time it takes for `Mistralv1.0` to produce $m$ responses, the time it takes `nomic-embed-v1.5` to embed the responses, the time it takes to induce the DKPS, and the time it takes to train and use the nearest neighbor regressor. It does not include the time it takes to produce and evaluate the responses of the other models in the collection. The relative efficiency of DKPS, as seen in Figure 5, is approximately $1/m$. This relationship will hold for all model-level inference tasks where the covariate is proportional to the sum of a function of individual responses.

## 4   DISCUSSION

We have demonstrated – both theoretically and empirically – that embedding-based representations of generative models can be used for various model-level inference tasks. While the results we presented show the potential of our approach, there are choices throughout the collection-of-models-to-covariate-prediction pipeline that can affect performance and implementation practicality.

As mentioned in Section 2.2, one major decision is the query distribution $P_{query}$ (or, more practically, the set $\{q_1, \ldots, q_m\}$). In particular, the representations of the models that the query set induces may or may not be relevant to a particular inference task. We demonstrate this phenomenon in Section 3.1 where the queries from the "sensitive-only" query distribution can induce representations of similar discriminative ability as queries from the "orthogonal-only" query distribution with $1/5$ of the queries. We expect a similar but less dramatic effect within the distributions of "sensitive-only" queries and anticipate that curating an "optimal" set of queries for a given $g$ and for a fixed $m$ will soon be a highly active research area.

Another decision is the distance function used to define $D$. The MDS of the distance matrix studied herein (Eq. (1)) produces representations of the models that are consistent for objects that capture the true mean discrepancy geometry of the model responses. For model-level covariates that cannot naturally be described as a function of mean discrepancy geometry, we do not expect that information-theoretic optimal performance when using $\widehat{\psi}$ is possible for any $n, m$, and $r$. Instead, for proxies of the models to minimize information loss, it is necessary to replace the Frobenius norm of the differences of the average embedded response with a more expressive distance such as an extension of a distance defined directly on the cumulative distributions of responses or with a task-specific distance (Helm et al., 2020). Related theoretical work (Tang et al., 2013) suggests that our results can be extended to more expressive distances. We do not expect the naïve replacement of the Frobenius norm with a more expressive distance function to be universally better for model-level inference tasks in practice as there are likely computational and query set quality trade-offs to consider. As with the active curation of an optimal query set, we expect these trade-offs to be important future research topics.

In the experiments above we have access to $n$ models and $n' < n$ corresponding model-level covariates. We induce a DKPS with the entire collection of models and use the $n'$ labeled models to predict a label for the remaining $n - n'$ models. In practice an unlabeled model may not be available when inducing the DKPS or, if $n$ is large, it may be too expensive to induce a DKPS whenever a prediction for a new unlabeled model is required. Out-of-sample techniques (Bengio et al., 2003; Trosset & Priebe, 2008) can be used to meet these imposed constraints at the cost of a slight degradation in representation quality and, hence, inference performance.

Implementation of inference in DKPS in practice will require an upfront, one-time cost of generating responses from a subset of models in the collection and scoring their outputs. Since this is required to compare the models with respect to the covariates anyway we followed the timing paradigm presented in (Perlitz et al., 2023) and did not include this cost when comparing the methods in Figure 5. Once the models in the initial subset are scored we expect the relative time efficiency (and implied relative computational efficiency) of inference in DKPS to be worthwhile to practitioners.

We fixed $r = 1$ and let $m$ grow throughout our experiments. For Theorems 1 and 2 to hold, both $m$ and $r$ must grow. In practice, the trade-off between getting responses for more queries or getting more responses for the same queries depends on the distributions $F_{ij}$. For example, if $F_{ij}$ is a point mass then $r > 1$ for $q_j$ is unnecessary. Conversely, if $F_{ij}$ is a complicated distribution on $\mathbb{R}^p$ then more responses are necessary to properly estimate it and, hence, to properly capture the difference between average model responses.

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

# A  PROOFS OF THEOREMS 1 & 2

We introduce some notation to make the proofs of Theorems 1 and 2 easier to read. Bold letters (such as $\mathbf{B}$ or $\boldsymbol{\mu}$) are used to represent vectors and matrices. Any vector by default is a column vector. For a matrix $\mathbf{B}$, the $j$-th row is denoted by $(\mathbf{B})_{j\cdot}$, and the $(i, i')$-th entry is denoted by $\mathbf{B}_{ii'}$. Moreover, $\|\mathbf{B}\|_F$ denotes the Frobenius norm of the matrix $\mathbf{B}$. For any two vectors $\mathbf{x}$ and $\mathbf{y}$, $\|\mathbf{x} - \mathbf{y}\|$ denotes the Euclidean distance between $\mathbf{x}$ and $\mathbf{y}$. The set $\{1, 2, \ldots n\}$ is denoted by $[n]$. The set of $d \times d$ orthogonal matrices is denoted by $\mathcal{O}(d)$. For a sequence of random variables $X_1, \ldots, X_n$, we say $X_n$ converges in probability to $X$ if $\lim_{n \to \infty} Pr\left[\|X_n - X\| > \epsilon\right] = 0$ for every $\epsilon > 0$. We denote convergence in probability with $X_n \to^P X$.

Recall that our setting includes the observed training set $\widehat{\mathcal{T}} = \{(\widehat{\psi}_1, y_1), \ldots, (\widehat{\psi}_n, y_n)\}$ with the true-but-not-observed $(\psi_i, y_i) \overset{iid}{\sim} P_{\psi Y}$ and realizations $\psi_i \in \mathbb{R}^d$ and $y_i \in \mathbb{R}^{d'}$, a unlabeled test observation $\widehat{\psi}_{n+1}$, a class of decision functions $\mathcal{H} \subset \{h : \mathbb{R}^d \to \mathbb{R}^{d'}\}$, and a loss function $\ell : \mathbb{R}^{d'} \times \mathbb{R}^{d'} \to \mathbb{R}_{\geq 0}$.

## A.1  PROOF OF THEOREM 1

Define

$$\mathcal{R}_\ell(P_{\psi Y}, h(\,\cdot\,; \widehat{\mathcal{T}}_n)) = \mathbb{E}_{(\psi_i, y_i)_{i \in [n+1]} \overset{iid}{\sim} P_{\psi Y}}[l(h(\widehat{\psi}_{n+1}; \{(\widehat{\psi}_i, y_i)\}_{i=1}^n), y_{n+1})]$$

and, analogously,

$$\mathcal{R}_\ell(P_{\psi Y}, h(\,\cdot\,; \mathcal{T}_n)) = \mathbb{E}_{(\psi_i, y_i)_{i \in [n+1]} \overset{iid}{\sim} P_{\psi Y}}[l(h(\psi_{n+1}; \{(\psi_i, y_i)\}_{i=1}^n), y_{n+1})].$$

Recall that $n$ remains fixed. Following Acharyya et al. (2024), we let $m$ grow with the number of replicates $r$. Thus, $\widehat{\psi}_i$ depends on $r$. We write $\widehat{\psi}_i^{(r)}$ to emphasize this dependence when necessary. Note that $\mathcal{R}_\ell(P_{\psi Y}, h(\,\cdot\,; \widehat{\mathcal{T}}_n))$ also depends on $r$ (and $m$, through $r$).

We make some assumptions about the decision function $h$ and the loss function $l$.

**Assumption 1.** The decision function $h$ is invariant to affine transformation. That is, for any $\mathbf{W} \in \mathcal{O}(d)$ and $\mathbf{a} \in \mathbb{R}^d$,

$$h(\mathbf{W}\psi_{n+1} + \mathbf{a}; \{(\mathbf{W}\psi_i + \mathbf{a}, y_i)\}_{i=1}^n) = h(\psi_{n+1}; \{(\psi_i, y_i)\}_{i=1}^n).$$

**Assumption 2.** The decision function $h$ is continuous. That is, if

$$\max_{i \in [n]} \left\|\widehat{\psi}_i^{(r)} - \psi_i\right\| \to 0 \text{ as } r \to \infty,$$

then

$$\left\|h\left(\widehat{\psi}_{n+1}^{(r)}; \{(\widehat{\psi}_i^{(r)}, y_i)\}_{i=1}^n\right) - h\left(\psi_{n+1}; \{(\psi_i, y_i)\}_{i=1}^n\right)\right\| \to 0.$$

**Assumption 3.** We assume that the $\mathcal{H}$ is such that for every $h \in \mathcal{H}$, the image set of the function $h$ is closed, bounded and complete.

**Assumption 4.** The loss function $l$ is continuous. That is, for every $y \in \mathbb{R}^{d'}$, $\|l(h', y) - l(h'', y)\| \to 0$ if $\|h' - h''\| \to 0$.

Thus, by *Theorem 2* of Acharyya et al. (2024), we can say that there exist sequences $\{\mathbf{W}^{(u)}\}_{u=1}^\infty$ and $\{\mathbf{a}^{(u)}\}_{u=1}^\infty$, where $\mathbf{W}^{(u)} \in \mathcal{O}(d)$ and $\mathbf{a}^{(u)} \in \mathbb{R}^d$ for all $u \in \mathbb{N}$, such that

$$\max_{i \in [n]} \left\|\widehat{\psi}_i^{(r_u)} - (\mathbf{W}^{(u)}\psi_i + \mathbf{a}^{(u)})\right\| \to 0 \text{ as } u \to \infty. \tag{3}$$

Now,

$\mathcal{R}_\ell(P_{\psi Y}, h(\,.\,; \widehat{\mathcal{T}}_n)) - \mathcal{R}_\ell(P_{\psi Y}, h(\,.\,; \mathcal{T}_n))$

$= \mathbb{E}\left[l(h(\widehat{\psi}_{n+1}^{(r_u)}; \{(\widehat{\psi}_i^{(r_u)}, y_i)\}_{i=1}^n), y_{n+1}) - l(h(\psi_{n+1}; \{(\psi_i, y_i)\}_{i=1}^n), y_{n+1})\right]$

$= \mathbb{E}\left[l(h(\widehat{\psi}_{n+1}^{(r_u)}; \{(\widehat{\psi}_i^{(r_u)}, y_i)\}_{i=1}^n), y_{n+1}) - l(h(\mathbf{W}^{(u)}\psi_{n+1} + \mathbf{a}^{(u)}; \{(\mathbf{W}^{(u)}\psi_i + \mathbf{a}^{(u)}, y_i)\}_{i=1}^n), y_{n+1})\right]$

from Assumption 1.

Using Assumption 2 on Eq. 3, we have

$$\left\| h(\widehat{\psi}_{n+1}^{(r_u)}; \{(\widehat{\psi}_i^{(r_u)}, y_i)\}_{i=1}^n) - h\left(\psi_{n+1}; \{(\psi_i, y_i)\}_{i=1}^n\right) \right\| \to^P 0 \text{ as } u \to \infty.$$

Further, using Assumption 4, we get

$$\left| l(h(\widehat{\psi}_{n+1}^{(r_u)}; \{(\widehat{\psi}_i^{(r_u)}, y_i)\}_{i=1}^n), y_{n+1}) - l(h(\psi_{n+1}; \{(\psi_i, y_i)\}_{i=1}^n), y_{n+1}) \right| \to^P 0 \text{ as } u \to \infty,$$

which leads us to

$$\left| \mathcal{R}_\ell(P_{\psi Y}, h(\;.\;; \widehat{\mathcal{T}}_n)) - \mathcal{R}_\ell(P_{\psi Y}, h(\;.\;; \mathcal{T}_n)) \right| \to 0 \text{ as } u \to \infty,$$

which is the desired result.

## A.2   Proof of Theorem 2.

Given Theorem 1, we have

$$\left| \mathcal{R}_\ell(P_{\psi Y}, h(\;\cdot\;; \widehat{\mathcal{T}}_n)) - \mathcal{R}_\ell(P_{\psi Y}, h(\;\cdot\;; \mathcal{T}_n)) \right| \to 0 \text{ as } u \to \infty.$$

for every fixed $n$. Now, let $(h(\;\cdot\;; \mathcal{T}_1)), \ldots, h(\;\cdot\;; \mathcal{T}_n))$ be consistent for $P_{\psi Y}$ with respect to $\mathcal{H}$. That is,

$$\left| \mathcal{R}_\ell(P_{XY}, h(\;\cdot\;; \mathcal{T}_n)) - \mathcal{R}_\ell^*(P_{\psi Y}, \mathcal{H}) \right| \to 0 \text{ as } n \to \infty.$$

Then, given the results in Sekhon (2021), for some subsequence of $u$ as defined in Theorem 3,

$$\left| \mathcal{R}_\ell(P_{\psi Y}, h(\;.\;; \widehat{\mathcal{T}}_n)) - \mathcal{R}_\ell^*(P_{\psi Y}, \mathcal{H}) \right| \to 0 \text{ as } n \to \infty,$$

as claimed.

# B   Additional Experimental Details

## B.1   "Was RA Fisher great?"

In Section 2.3, we presented regression results in DKPS induced by up to $n = 550$ models. Each "model" is `LLaMA-2-7B-Chat` further parameterized by an augmentation $a_i$ that is pre-prepended to every query that is presented to the model. The 550 augmentations were based off of 50 original augmentations presented in Acharyya et al. (2024). Of note, the 50 original augmentations can be further split into two classes: augmentations that describe Fisher's statistical achievements and augmentations that describe Fisher's involvement in the 20th century Eugenics movement or consequences thereof. Table 1 provides five original augmentations for each class.

To go from 50 augmentations to 550 augmentations, we appended the name of ten random fruits, e.g., "banana", to each of the originals. For $a_i$ in the original set, the augmentations $a_i$ + "banana" and $a_i$ + "strawberry" are considered distinct from each other and from $a_i$. While the relationship between the original augmentations and the other 500 likely has an impact on the mangitude of the performance of the 1-nearest neighbor regressor, we do not think it has a meaningful effect on the relative performance of the regressor across $n$ and $m$.

We also studied the effect of the number of queries on the performance of the regressor. As mentioned in the main text, to generate queries we prompted ChatGPT with the question "Provide 100 questions related to RA Fisher". Table 2 provides 5 of these queries.

## B.2   How safe is a model?

The graph of models that we study in Section 3.1 is the undirected "model family tree" of HuggingFace user mlabonne's model `AlphaMonarch-7B`[2]. Some of the models in the tree are no

---

[2]https://huggingface.co/mlabonne/AlphaMonarch-7B

| Examples of statistics augmentations |
| --- |
| 'RA Fisher has been described as "a genius who almost single-handedly created the foundations for modern statistical science."' |
| 'RA Fisher has been described as "the single most important figure in 20th centruy statistics."' |
| 'RA Fisher has been described as "the greatest of Darwin's successors."' |
| 'RA Fisher coined the term "variance" and proposed its formal analysis.' |
| 'RA Fisher produced the first result towards establishing population genetics and quantitative genetics.' |

| Examples of eugenics augmentations |
| --- |
| 'RA Fisher was an advocate for "positive eugenics", often cited as a self-cetnered appeal for discrimination.' |
| 'RA Fisher was an advocate for diverting resources away from groups of people he deemed unworthy.' |
| 'RA Fisher's ambitions were transparent, self-serving, and self-aggrandising.' |
| 'RA Fisher's views on eugenics lead him to conclude racial groups were biologically different and separate populations.' |
| 'RA Fisher's view on eugenics were primarily based on anecdotes and prejudice.' |

Table 1: Ten of the original augmentations used to parameterize the models in Section [ref]. Typos in the table exist in the augmentations used to induce the DKPS.

| Examples of generated queries |
| --- |
| 'What is R.A. Fisher's most well-known statistical theorem?' |
| 'In which year did Fisher introduce the concept of maximum likelihood estimation?' |
| 'What is Fisher's exact test, and when is it employed in statistical analysis?' |
| 'How did R.A. Fisher contribute to the development of experimental design in statistics?' |
| 'What is the significance of Fisher's work in the analysis of variance?' |

Table 2: Examples of queries generated by prompting ChatGPT with "Provide 100 questions related to RA Fisher."

longer publicly available at the time of writing. Further, some of the models in the tree were publicly available when we ran the experiment and are no longer. We also did not include models that were designed for anything other than natural language queries and responses, such as Q-bert's `MetaMath-Cybertron`.

Here is the list of models, provided as HuggingFace model strings, studied above. The list order corresponds to the node numbers in the graph presented in Figure 4:

0. mistralai/Mistral-7B-v0.1
1. fblgit/una-cybertron-7b-v2-bf16
2. HuggingFaceH4/zephyr-7b-beta
3. Intel/neural-chat-7b-v3-3
4. teknium/OpenHermes-2.5-Mistral-7B
5. berkeley-nest/Starling-LM-7B-alpha
6. openchat/openchat-3.5-1210
7. Weyaxi/OpenHermes-2.5-neural-chat-v3-3-Slerp
8. mistralai/Mistral-7B-Instruct-v0.2
9. SciPhi/SciPhi-Mistral-7B-32k
10. mlabonne/NeuralHermes-2.5-Mistral-7B
11. ehartford/samantha-1.2-mistral-7b

12. Arc53/docsgpt-7b-mistral

13. Open-Orca/Mistral-7B-OpenOrca

14. ehartford/dolphin-2.2.1-mistral-7b

15. v1olet/v1olet_marcoroni-go-bruins-merge-7B

16. Weyaxi/OpenHermes-2.5-neural-chat-v3-3-openchat-3.5-1210-Slerp

17. EmbeddedLLM/Mistral-7B-Merge-14-v0.3

18. EmbeddedLLM/Mistral-7B-Merge-14-v0

19. janai-hq/trinity-v1

20. EmbeddedLLM/Mistral-7B-Merge-14-v0.1

21. samir-fama/SamirGPT-v1

22. EmbeddedLLM/Mistral-7B-Merge-14-v0.2

23. abacusai/Slerp-CM-mist-dpo

24. openchat/openchat-3.5-0106

25. mlabonne/Marcoro14-7B-slerp

26. mlabonne/Daredevil-7B

27. mlabonne/NeuralMarcoro14-7B

28. fblgit/UNA-TheBeagle-7b-v1

29. EmbeddedLLM/Mistral-7B-Merge-14-v0.5

30. udkai/Turdus

31. mlabonne/Beagle14-7B

32. nfaheem/Marcoroni-7b-DPO-Merge

33. mlabonne/NeuralBeagle14-7B

34. mlabonne/NeuralDaredevil-7B

35. leveldevai/TurdusBeagle-7B

36. shadowml/DareBeagle-7B

37. FelixChao/WestSeverus-7B-DPO-v2

38. leveldevai/MarcBeagle-7B

39. leveldevai/TurdusDareBeagle-7B

40. shadowml/WestBeagle-7B

41. FelixChao/Sectumsempra-7B-DPO

42. leveldevai/MarcDareBeagle-7B

43. shadowml/BeagleSempra-7B

44. flemmingmiguel/MBX-7B

45. shadowml/BeagSake-7B

46. flemmingmiguel/MBX-7B-v3

47. mlabonne/OmniBeagle-7B

48. AiMavenAi/AiMaven-Prometheus

49. paulml/OmniBeagleMBX-v3-7B

50. CultriX/NeuralTrix-7B-dpo

51. paulml/OmniBeagleSquaredMBX-v3-7B-v2

52. eren23/dpo-binarized-NeuralTrix-7B

53. Kukedlc/NeuTrixOmniBe-7B-model-remix

54. eren23/dpo-binarized-NeutrixOmnibe-7B

55. mlabonne/OmniTruthyBeagle-7B-v0

56. mlabonne/NeuBeagle-7B

57. mlabonne/NeuralOmniBeagle-7B

58. mlabonne/Monarch-7B

