# OpenReview forum: "Embedding-based statistical inference on generative models"
_ICLR.cc/2025/Conference — ICLR 2025 Conference Withdrawn Submission_

### Official Review · Reviewer_nufo · 2024-10-21

**Soundness:** 2
**Presentation:** 2
**Contribution:** 2
**Rating:** 3
**Confidence:** 4

**Summary:**

The paper titled "Embedding-Based Statistical Inference on Generative Models" presents methods to leverage embedding-based representations of generative models for statistical inference tasks. These models, widely recognized for their expert-level content generation across domains, can vary significantly based on covariates such as benchmarks or model properties. The paper proposes a method to predict these covariates when direct access is unavailable.

The core contribution is the extension of embedding-based representations, specifically the "data kernel perspective space" (DKPS), into classical statistical inference. This allows users to infer model-level properties by using the embeddings of similar models. The authors demonstrate this through various tasks, such as predicting the presence of sensitive information or a model’s safety.

**Strengths:**

The paper is grounded in strong theoretical foundations, drawing from recent advancements in embedding-based representations and applying them to real-world inference problems. The extension of prior work on model embeddings to the statistical inference setting is supported by well-reasoned arguments and formal proofs. Empirical evaluations across multiple tasks, such as model safety and sensitivity analysis, reinforce the claims made, with results that demonstrate both the effectiveness and scalability of the proposed method. The experimental design appears sound, and the methodology is reproducible based on the description provided.

**Weaknesses:**

The paper claims to introduce a novel framework by extending embedding-based representations to infer model-level covariates, but this concept is only a modest extension of existing techniques. Embedding-based methods are already widely applied for clustering, classification, and regression tasks. The application to generative models may appear new, but it is essentially an adaptation of standard techniques rather than a breakthrough innovation.

The experiments are narrow in scope, and the datasets or tasks chosen do not convincingly demonstrate the real-world applicability or generalizability of the proposed methods. While the toy example (predicting if a model will say "yes" to "Was RA Fisher great?") provides an illustration, it lacks complexity and doesn't reflect more challenging, realistic tasks. The experimental design also fails to explore the scalability of the approach or its robustness to different hyperparameters or model sizes.

**Questions:**

How does your approach scale to larger models or model collections, especially when working with large-scale, real-world generative models like GPT-4 or other multi-billion parameter models? Did you consider any strategies to optimize or manage the computational cost in such settings?

The tasks chosen for your experiments, such as predicting the response to "Was RA Fisher great?" or detecting sensitive information, seem narrow and possibly not reflective of more complex model inference challenges. How do you justify the choice of these tasks, and do you plan to expand the scope of the evaluation in future work?

You provide results on performance improvements using DKPS, but there is a lack of direct comparison with alternative approaches. Could you explain why certain baselines (e.g., traditional statistical inference methods or other embedding-based inference techniques) were not included for comparison, and how you plan to address this in future work?

There is limited discussion on the sensitivity of your method to hyperparameters such as the number of queries (m) or the number of models (n) used in generating the DKPS. How sensitive is your approach to these choices, and what strategies do you recommend for optimizing these parameters in practice?

In the paper, you mention that DKPS can predict various covariates, but it's unclear how well the approach generalizes across different types of covariates (e.g., safety, hallucination rate, bias). Do some covariates work better than others with DKPS, and are there any limitations on the kinds of covariates that can be inferred using your method?

How does your method handle models of varying complexity, especially when combining simpler models with highly complex generative models in the same DKPS? Does the variance in model complexity affect the performance of the DKPS?

---

> ### Author Response · Authors · 2024-11-14
>
> nufo,
>
> Thank you for taking the time to read and review our paper. We have addressed your part of your review below.
>
> $ \textbf{Summary}$ :
>
> We think your summary is a fair description of our work.
>
> $ \textbf{Strengths}$:
>
> We agree with your noted strengths of the paper.
>
> $ \textbf{Weaknesses}$:
> > The paper claims ...
> > The experiments are narrow ...
>
> We agree that our work is an extension of existing embedding-based analysis. We disagree that it is only a modest extension. To go from analyzing the embeddings of a collection of documents to analyzing the embeddings of generative models requires 1) an aggregation over responses, 2) an aggregation over queries, and 3) a suitable metric on the aggregation over queries. There are a number of options for each of these three steps -- which opens the door for theoretical and empirical explorations of numerous instances of similar methods -- that suggests, to us, that the extension is not trivial.
>
> Our goal with the experiments was to demonstrate the general application of the proposed method. For example, we consider one classification task and three regression tasks and consider three different collections of models across the four tasks -- one collection parameterized by different fixed augmentations, one collection parameterized by different LoRA matrices, and one collection of “off-the-shelf” but related models. On one hand, this demonstrates broad applicability. On the other hand, our analysis was limited to 7B models and quantized versions of 7B models due to very real computational and budget constraints. While we expect our results to hold for collections of larger models, we agree that including such an example would improve our paper.
>
> $ \textbf{Questions} $:
> > How does your approach scale ...
>
> Our method is quite general and can be applied to larger models or model collections. With that said, one of the components of our method is querying each model multiple times. We expect that this is unavoidable in the black-box setting that we address -- though we are actively working on ways to get high-quality representations of the models in as few queries as possible. Some promising directions that we are working on include selecting a core-set of queries, developing an “optimal” metric on the aggregation of queries, and ensembling across embedding functions.
>
> > The tasks chosen ...
>
> Outside the context of benchmarking, statistical analysis of populations of models is relatively new. As such, there are relatively few standard tasks. We took this as an opportunity to explore what we thought were interesting and relevant tasks.
>
> Given that the treatment of our setting as a classical supervised learning problem is relatively new, the choice of using the “Was RA Fisher great?” as our leading example was mainly due to its simplicity. The covariate y = P[“yes” | augmentation] is easy to understand. Further, the model-level covariate surface should correlate strongly along the dimension in the perspective space that captures difference in the eugenics and statistics augmentations.
>
> We disagree that “detecting [if a model has / had access to] sensitive information” task is narrow and not reflective of more complex model inference challenges. See, e.g., this popular press article https://www.cnbc.com/2024/03/06/gpt-4-researchers-tested-leading-ai-models-for-copyright-infringement.html.
>
> Our sensitive data example is a toy example that is similar-in-kind to detecting if copyrighted information was in the pre-training / finetuning data of a black-box model, if private or personal information is in the pre-training / finetuning data of a black-box model, etc. We’d argue that this is one of the major legal and ethical data security & privacy problems of the current technological era.
>
> > You provide results ...
>
> We are not aware of any black-box methods that are capable of addressing our setting. We’d be happy to include comparisons if you could direct us to the methods that you have in mind!
>
> NB: The response to the rest of your review is provided in a follow on comment.

---

> > ### Author Response · Authors · 2024-11-14
> >
> > This is the continuation of our response to nufo's review.
> >
> > > There is limited discussion ...
> >
> > Great question! As mentioned above, along with the number of queries (m) and the number of models (n), there are numerous other hyperparameters to consider when generating the DKPS, such as how to aggregate responses, how to aggregate over queries, and the metric on the aggregations over queries. It is also important to consider the embedding function (g), the distribution of queries (G), and the number of replicates per query (r).
> >
> > We have found that the two most impactful hyperparameters are the embedding function (g) and the distribution of queries (G). The impact of these are relatively intuitive. For example, if the embedding function that you use is not capable of capturing the inherent difference between the models, then the DKPS representations will not be reasonably separated. Similarly, if the queries that you are using to prompt the model do not allow for the models to express their differences, then the DKPS representations will not be reasonably separated.
> >
> > We explored the effect of different distribution of queries in our sensitive-data classification experiment and showed that queries that are relevant to the difference in model classes can significantly reduce the number of required queries to achieve a particular performance. We did not explore the effect of different embedding functions in this paper since its effects on more common problems (such as classification, clustering, etc.) documents is well-known.
> >
> > There is a lot to explore with respect to this method -- which we discuss in the Discussion -- and we are excited to continue to push it forward.
> >
> > > In the paper, you mention ...
> >
> > Great question! The representations of the models in DKPS can be used to perform any inference task on the models themselves. In this sense, DKPS can be used to predict any type (real valued, categorical, etc.) of covariate. How well-suited the representations are for a particular task will depend on the hyperparameters discussed above.
> >
> > We discuss this in the paragraph that starts on line 200/201 (“While we do not ..”). In summary, if the covariate can be described as a function of the average of the model responses then the current DKPS pipeline can be used for optimal performance. If, instead, the covariate is a more complicated function of model responses then a variation of the proposed method should be used.
> >
> > We tried to address model safety in the model toxicity & bias experiment -- please let us know how we could better address “safety”!
> >
> > > How does your method ...
> >
> > Great question -- since the foundation of the DKPS is a collection of embeddings associated with each model, our method would be affected by response complexity for a given set of queries, not necessarily “model” complexity. With that said, the proportion of models with a particular level of response complexity is bounded below by some \epsilon > 0 as the number of models grows, then the performance will not be meaningfully affected.
> >
> > Please let us know if there are anything else we can clarify or any other questions you may have throughout this discussion phase.

---

### Official Review · Reviewer_akqW · 2024-11-03

**Soundness:** 2
**Presentation:** 2
**Contribution:** 3
**Rating:** 5
**Confidence:** 4

**Summary:**

The submission proposes an embedding-based approach for statistical inference within generative models. Leveraging the Data Kernel Perspective Spaces framework, the proposed method allows to embed models into shared geometric space based on models' responses to a set of queries. These responses are embedded with an external embedder and are averaged. The distance matrix between the models in the collection is then formed based on the differences in the averaged embedded responses. The model embedding is then obtained by applying a multidimensional scaling on the obtained distance matrix. Thus, MDS provides the d-dimensional vector representations for the collection of models called data kernel perspective space.

The submission also offers consistency results for the proposed approach, which essentially say that if one increases the number of queries, learning on the observed data kernel perspectives (model embeddings) will be close enough to learning on true unknown perspectives.

To enable analysis and model property inference, the approach requires a careful design of the query set and a set of labelled models which could either be obtained from external sources (scores from benchmarks) or fine-tuning models on the controllable training sets to form a reference model set with labels. By analysing distances and configurations within DKPS, one can infer similarities and differences in model behaviours, even if these models differ in architecture, training data, or parameters. The subsequent empirical study explores the viability of this approach.

**Strengths:**

The idea of substituting the direct model evaluation on existing benchmarks with an embedding approach and subsequent examination of relations within a collection of models in this space seem novel and more general in terms of applicability and computational efficiency.
This is especially important in scenarios where score function for assessing the model output is not formalised or unavailable (e.g., there is no respective benchmark).

Theoretical guarantees for the consistency of DKPS representations and further empirical validation are provided to ensure results are stable and reliable with increasing amount of query and model set sizes.

The computational efficiency makes DKPS a practical tool for large-scale applications where many generative models need to be evaluated and compared quickly.

In terms of clarity, the paper does a good job describing the approach and providing empirical illustration.
The authors also discuss limitations in terms of design choices for particular pipeline implementations, involving choice of query set formation and distance functions used in MDS.

**Weaknesses:**

1. Distinction with prior work is not clear to me. After a quick glance at [1], which current submission heavily relies on (DKPS, consistent representations, the empirical illustration with RA Fisher greatness, ablations), it is difficult to distinguish the contributions of the current paper because distinctions are not clearly stated. I would appreciate if authors could specify how their current work departs from the previous research.

2. The toxicity and bias experiments are not entirely clear to me. The model's covariate correlates with the size of the point in Figure 4 (right, top), which demonstrates that DKPS doesn't provide a simple configuration for these particular covariates unlike the previous example in Figure 2. Surely, this is expected, since both bias and toxicity are of a more complex nature. But this also highlights that limitation of the proposed approach. Although, the 1-NN in DKPS space for the particular example (green, blue and red models) shows benefit over other simple regressors, it would be more convincing to see this result at scale. As far as I understood, Figure 4. (right, bottom) is also computed for one unlabelled model and underlines the hardness of the problem.

3. The proposed approach highly depends on the design choices and it is not explored how much effort is required to produces pipelines that are descriptive and practical enough. If this work positions itself as a practical extension of the theoretical framework introduced in prior work, the undeniable contribution would be to provide a convincing practical application. For example, although the existing experiment with toxicity and bias helps understand that green model is closer to less toxic red model than the more toxic blue model, this doesn't answer the question whether it can be less toxic than the red one, which is probably the question practitioners would be interested in. Finding a better scenario to showcase the strength of the proposed approach, would help this submission greatly.

[1] Acharyya, Aranyak, et al. "Consistent estimation of generative model representations in the data kernel perspective space." arXiv preprint arXiv:2409.17308 (2024).

**Questions:**

1. How this work differs from [1], which seems to introduce the DKPS and provides similar empirical study?
2. Did I understand it correctly, that Figure 4 (right, bottom) consider 1 unlabelled model? I suggest demonstrating this at scale, meaning testing performance of the proposed approach on at least several unlabelled models to find whether the results hold.
3. Is my understanding right that we cannot see the clear dependence in the DKPS spaces with the covariates in Figure 4 right top because the complex nature of the covariates? Might this mean the query sets are not optimal for the covariates to demonstrate variation along one of the 2 top dimensions?
4. On line 112, do you mean $j$th row in $\mu_i$?

Addressing the distinction with prior work, providing more comprehensive empirical study (showing results hold for many models), and demonstrating a more practical setup where DKPS alleviates the need for direct model evaluation on a benchmark would help raise the score for this very promising paper!

---

> ### Author Response · Authors · 2024-11-14
>
> akqW,
>
> Thank you for taking the time to read and review our paper. We have addressed your review below.
>
> $ \textbf{Summary} $:
>
> We think your summary is a fair description of our work.
>
> $ \textbf{Strengths} $:
>
> We agree that it is important to develop black-box methods to address model-level inference in scenarios where the score function is not formalized or unavailable. We think that the level of importance is still growing -- especially with large corporations putting retrieval-enabled language models on user’s phones, tablets, etc. and the ongoing pursuit of "digital twins" of society with the current cohort of language models.
>
> $ \textbf{Weaknesses}:
>
> > Distinction ...
>
> The distinction between the current submission and [1] is that we analyze the representations of the models in the estimated DKPS with respect to a subsequent inference task. The results of [1] essentially say that the low-dimensional representations of the models in the DKPS are consistent for a true-but-unknown set of low-dimensional representations of the models. They do not have a notion of model-level covariates. As such, they do not explicitly comment on the utility of the representations or on the convergence of decision rules.
>
> > The toxicity and bias ...
>
> Yes, the model’s covariate correlates with the size of the point. & yes, the nature of bias and toxicity is more complicated than the other two examples. One of the goals of the bias/toxicity example was to demonstrate that functional difference (as measured by “closeness” in DKPS) is more important than model lineage (as measured by “closeness” in the model graph), even when the relationship between the DKPS and the covariate is non-linear. Also, while we only highlight a single set of neighbors in the model graph and the two DKPS, the relative error shown in the bottom right of Figure 4 is actually the average relative error over all models in the model tree. We will make sure to make that more clear. We will also emphasize the comparison of “functional” versus “lineage”.
>
> > The proposed approach ...
>
> Our goal is to be a practical extension of the more theoretical [1], and so we greatly appreciate your suggestions. Beyond the illustrative example, we tried to pick two interesting experiments demonstrating practicality and attempted to highlight potential impact of important hyperparameters. For example, we explored sensitivity to an important hyperparameter (the distribution of queries G) to show that consistency of the estimated DKPS is *not* the only important thing, as one might think after reading [1].
>
> We think of the sensitive data classification example as a simple experiment related to predicting if a model has had access to copy-righted or private information, which we think is one of the biggest ethical and legal questions related to training and using generative models. See, for example, this popular press article: https://www.cnbc.com/2024/03/06/gpt-4-researchers-tested-leading-ai-models-for-copyright-infringement.html.
>
> Related to the question of the blue model being more (or less) toxic than the red model, we agree that this is an important question. We expect that one use case for DKPS will be to quickly identify not-passable models so that resources can be better used to evaluate promising models. For example, we could safely remove models from consideration if they are close to the more toxic blue model and only spend evaluation resources on models close to the less toxic red model.
>
> $ \textbf{Questions} $:
>
> >How this work ...
>
> [1] introduces DKPS and focuses entirely on the behavior of the estimate of the DKPS. It shows that the estimated DKPS is consistent for a true-but-unknown geometry of models. [1] does not try to demonstrate the utility of the estimated DKPS with respect to model-level inference. As such, we extend the theoretical results to include inference and (attempt to) show that the DKPS is broadly applicable to many model-level problems.
>
> > Did I understand ...
>
> Figure 4 (right, bottom) shows the average performance of the classifiers across all models in the model graph.
>
> > Is my understanding ...
>
> There are many reasons why we might now see clear dependence in the DKPS space. One might be because two dimensions is not enough to capture it perfectly. Others, as you pointed out, may be due to the choice in hyperparameters such as the query set, the embedding function, the choice of metric on the models, etc. With that said, the shown DKPS is still better than other simple methods for trying to predict toxicity / bias, such as using the model graph or using the global average. This highlights that while the covariate signal in the DKPS is not necessarily linear, there is still a signal in the shown DKPS geometry.
>
> > On line 112 ...
>
> Yes! Great catch, thank you!
>
> > Addressing ...
>
> Thank you for your helpful feedback! Please let us know if our responses were helpful in understanding our work or if there are any other questions you have.

---

> ### Comment · Reviewer_akqW · 2024-11-26
> **thank you for response, concern on empirical study and method's foundation paper, generative model comparison baselines**
>
> Dear authors, thank you for a detailed response! My questions are answered, but as mentioned earlier, more comprehensive empirical study is needed (though I understand that there is no time left to conduct larger scale experiments). For example, testing the method's predictive capacity with model embeddings of higher dimensions in other settings (not bias and toxicity, but more quantifiable ones like standard NLP benchmarks) would help validate the applicability of the method.
>
> After reading the other reviews, I agree that the background literature is very recent and [1] has not been published yet, raising concern. There is another concern on the baselines raised by reviewer **nufo**, there are several model comparison methods that offer model embedding perspective as well, e.g. [1-3] with KID the closest in spirit to DKPS method.
>
> [1] Bińkowski, Mikołaj, et al. "Demystifying mmd gans." arXiv preprint arXiv:1801.01401 (2018).
>
> [2] Khrulkov, Valentin, and Ivan Oseledets. "Geometry score: A method for comparing generative adversarial networks." International conference on machine learning. PMLR, 2018.
>
> [3] Tsitsulin, Anton, et al. "The shape of data: Intrinsic distance for data distributions." arXiv preprint arXiv:1905.11141 (2019).

---

> > ### Author Response · Authors · 2024-11-27
> >
> > akqW,
> >
> > Thank you for taking the time to read and respond to our response! We appreciate your feedback and understand that more empirical studies would help validate the applicability of the method.
> >
> > The three papers you shared compare the geometries of data for pairs of generative models -- not entire collections of models -- and do not attempt to define or study inference on the models themselves. We want to emphasize that our paper is not an attempt to investigate the differences in the geometries of data from the perspectives of different models directly, it is an attempt to characterize and understand the geometry of the models themselves. While one could use the measures of (dis)similarity on geometries of data studied in the three papers you shared as the way to calculate entries of the pairwise distance matrix in the DKPS pipeline (our Eq. 1), there are no reasons to prefer theirs over ours $ \textit{a priori} $.
> >
> > As we mentioned in our response to $ \textbf{nufo} $ we do not know of any black-box methods that address our setting or problem.

---

### Official Review · Reviewer_vySb · 2024-11-04

**Soundness:** 2
**Presentation:** 1
**Contribution:** 2
**Rating:** 3
**Confidence:** 3

**Summary:**

The paper proposes a framework for performing statistic inference on collections of generative models with a particular focus given empirically to language models. The approach is defined through a "data kernel perspective space" and aims to statistically evaluate decision functions over this space.

**Strengths:**

The general problem space is of broad interest to the community at large, with quantifying various aspects of ever-growing foundation / generative models. Any effort towards making these models less black-box-like is commendable, especially given how few assumptions this approach makes at the outset.

**Weaknesses:**

I found the work to be trying to be too general at the cost of messaging and clarity. Throughout the detailing of the methodology, it is unclear what the precise goal is when trying to construct this statistical inference framework for generative models. I understand the general intent to perform inference in the presence of multiple different models, different generations, and different settings; however, in covering so much it is not obvious if anything meaningful can come out of so many degrees of freedom in the setup. The empirical findings are not too convincing to me, as the theory developed does not appear to tie too closely to the results. For example, in Figure 1, could be equivalently achieved with a logistic regression model trained on the same embeddings. Any results concerning actual statistical tests appear to be from other established tests, executed after the initial embedding in this "data kernel perspective space".

Additionally, I feel that in striving to make this approach generic and applicable to many different settings, the text became fairly unreadable and indecipherable to me. The mix of background on the data kernel perspective space mixed with adapting it to a generic statistical inference framework introduced a lot of notation in what I found to be a very terse manner.

Finally, I have some general concerns about the soundness of the foundations that this work builds upon. In section 1.1, a great deal of deference is made to previous works that this paper builds off of. In particular, the three listed below, with specific emphasis on the first. This paper claims to develop further results in this line of work, it does appear to heavily rely on the theoretical findings of these previous ones. I bring this up because all of these papers are preprints that have come out in the past 1.5 years (two in the past 6 months), with absolutely no citations (except between themselves) and no peer review. For more empirical-leaning work, this wouldn't cause me much concern but I feel for such a theoretically positioned paper, it is important that the foundation being built upon is trustworthy.
1. Acharyya, A., Trosset, M. W., Priebe, C. E., & Helm, H. S. (2024). Consistent estimation of generative model representations in the data kernel perspective space. arXiv preprint arXiv:2409.17308.
2. Helm, H., Duderstadt, B., Park, Y., & Priebe, C. E. (2024). Tracking the perspectives of interacting language models. arXiv preprint arXiv:2406.11938.
3. Duderstadt, B., Helm, H. S., & Priebe, C. E. (2023). Comparing Foundation Models using Data Kernels. arXiv preprint arXiv:2305.05126.

**Questions:**

No additional questions, please address my concerns above. In general, if I am misunderstanding the parts or all of this paper, please do let me know as I could very well have missed something critical in my reading.

---

> ### Author Response · Authors · 2024-11-14
>
> vySb,
>
> Thank you for taking the time to read and review our paper. We have addressed your review below.
>
> $ \textbf{Summary} $:
>
> We think your summary is a fair description of our work.
>
> $ \textbf{Strengths} $:
>
> We agree that trying to understand various aspects of the ever-growing set of generative models is extremely important. Our goal was to abstract away a lot of the different types of differences in models (such as model weights, augmentation strategies, generation parameters, etc.) in a single, unified framework.
>
> $ \textbf{Weaknesses} $:
> > I found the work to be trying to be too general ...
>
> Thank you for this feedback. One of our goals in writing this paper is to establish a general approach to thinking about model-level analysis and inference. Our method -- inference in the data kernel perspective space (DKPS) -- allows for classical statistical inference (classification, regression, etc.) of Euclidean representations of the models. Since we treat the models as black boxes, the method is agnostic to different types of model differences such as model weights, generation settings, or augmentation strategies. While we may sacrifice depth for breadth, we think it is important to make sure that the reader is aware that the DKPS framework can handle a variety of types of model differences.
>
> Our contribution is not intended to be that a particular decision function (1-NN, LDA, etc.) is the best for a particular task. It is to demonstrate that the model representations that are presented in Acharrya et al. can be used in the context of model-level inference. As such, we agree that a logistic regression model trained on the same model-level embeddings (from DKPS) would be effective for the “Was RA Fisher great?” task.
>
> Our theoretical results are very related to our empirical results. In particular, Theorem 1 states that as we get more queries (m) the performance of any classifier gets closer to the performance of the classifier trained on the true representations of the models. Indeed, we see that an increase in m corresponds to an increase in performance across all of the experimental settings. Similarly, Theorem 2 states that if a sequence of decision rules trained on the true-but-unknown representations of the models is consistent then the analogous sequence of decision rules trained on the estimated representations of the models is also consistent. This explains why we see performance improvements as a function of the number of models (n) in both the “Was Fisher great?” and “Has a model seen sensitive data” examples.
>
> We are a bit confused by your comment related to “results concerning actual statistical tests appear to be from other established tests, executed after the initial embedding in this [DKPS]”. As mentioned above, our contribution is *exactly* that inference in the DKPS is meaningful. We show this in classification settings, regression settings, and hypothesis testing settings. We think that this is an extremely important contribution, as there are currently no existing methods of getting model representations (that we know of, please point us to them if they exist!) that achieve this type of utility for populations of black-box models.
>
>
> > Additionally ...
>
> We plan to edit the background and generic statistical inference framework sections to be more reader friendly. Any suggestions that would help make these sections more readable would be greatly appreciated.
>
> > Finally, I have some general concerns about the soundness of the foundations ...
>
> This is a super fair and valid concern.Tracking perspectives of interacting language models has recently been published at EMNLP. Consistent estimation of generative model representations in the data kernel perspective space is currently under review at a dedicated statistics venue. This field is quite new and progress is fast-paced.
>
> With that said, the current work builds upon more established theoretical work related to inference in learned node embedding spaces of random graphs. See, e.g., [1] and [2].
>
> [1] Minh Tang. Daniel L. Sussman. Carey E. Priebe. "Universally consistent vertex classification for latent positions graphs." Ann. Statist. 41 (3) 1406 - 1430, June 2013. https://doi.org/10.1214/13-AOS1112
> [2] Avanti Athreya, et al. “Statistical inference on random dot product graphs: a survey” Journal of Machine Learning Research 18 (2018) 1-92.
>
> $ \textbf{Questions} $:
>
> Thanks again for taking the time to read and review our work. We recognize that we attempt to do a lot in the allotted pages and hope that discussions with you will help improve the presentation of our work.
>
> Please let us know if there is anything else we can clarify or you’d like to discuss.

---

> > ### Comment · Reviewer_vySb · 2024-12-03
> >
> > Thank you for clarifying some points and replying to my review. After reading the other discussions and some contemplation, I still do not think that current paper and the additions are fit for publication. This is an important line of work and I encourage the authors to further refine their message in the future. As of this time, I will not be changing my original score (3).

---

### Official Review · Reviewer_9WTf · 2024-11-04

**Soundness:** 1
**Presentation:** 1
**Contribution:** 2
**Rating:** 3
**Confidence:** 2

**Summary:**

Authors consider embedding-based representations of generative models and demonstrate that it can be used for several model-level inference tasks (predicting whether or not a model has had access to sensitive information and predicting model safety).

**Strengths:**

As the landscape of generative large language models evolves, it is important to develop new techniques to study and analyze differences in model behavior. The authors show the potential of embedding-based vector representations for capturing meaningful differences in model behavior in the context of a set of queries.

**Weaknesses:**

1. The authors note that their work extend theoretical results of the paper "Consistent estimation of generative model representations in the data kernel perspective space" (arxiv, 2024; Aranyak Acharyya, Michael W. Trosset, Carey E. Priebe, and Hayden S. Helm). The literature review in sec 1.1 highlighted the following works as of particular relevance: "Comparing foundation models using data kernels" (arxiv, 2023; Brandon Duderstadt, Hayden S Helm, and Carey E Priebe) and "Tracking the perspectives of interacting language models." (arxiv, 2024; H Helm, B Duderstadt, Y Park, CE Priebe). I am concerned that the referenced works are unpublished preprints that are not cited by other works. This may make it difficult for the reader to assess the significance of the research and trace its continuity.

2. I can't understand the first paragraph of section 2. What does index "i" mean, what does index "r" mean?

3. The second paragraph in section 2 contains exactly two sentences that are not related to each other in any way.

4. Line 102: "is the average difference between the average embedded response between" - it's hard to understand.

5. Lines 189-192. The text is difficult to understand, perhaps it should be rephrased and the multiple repetitive "that" removed. Why is the word "optimal" in quotation marks here? In what sense do you understand optimality here?

6. In general, from the text after theorems 1 and 2, at first glance it looks like nothing at all follows from the theorems ("The result does not provide instructions";  "Nor does it provide insight"; "does not provide guidance"; "it is unclear how"). Perhaps this text should be rephrased and it should be more clearly outlined what follows from these theorems and how you use them further in the work.

7. Line 796: "we appended the name of ten random fruits, e.g., “banana” ... ". I don't understand how fruits came to be here, perhaps the authors should explain this.

Overall, it seems to me that the work is very relevant, but before presenting it at the conference, the text and style of presentation should be significantly revised. Therefore, I cannot recommend this work for acceptance.

**Questions:**

Please see above.

---

> ### Author Response · Authors · 2024-11-14
>
> 9WTf,
>
> Thank you for taking the time to read and review our paper. We have addressed your review below.
>
> $ \textbf{Summary} $:
>
> We think your summary is a fair description of our work.
>
> $ \textbf{Strengths} $:
>
> We agree that it is important to develop new techniques to analyze differences in model behavior -- or, more generally, to understand and study populations of models.
>
>
> $ \textbf{Weaknesses} $:
>
> > The authors note ...
>
> That is a valid and fair concern. “Tracking perspectives of interacting language models” has since been accepted to EMNLP. The theoretical work that our paper extends is currently under review at a dedicated statistics venue. The study of populations of language models is quite new and, as such, there has been quite a lot of fast-paced progress.
>
> > I can't understand ...
>
> $ f_{i} $ is one of the n models of the previous sentence. In general, we let $ i $ & $ i’ $ index models and $ j $ and $ j’ $ index queries.
>
> $ r $ represents the number of responses from a given model for a particular query. For example, if we fix the model to be GPT4 and the query to be “What is your favorite color?” then $ f_{i} (q_{j})1, ..,  f_{i}(q_{j})r $ could be “red”, “red”, “blue”, .., “red”. This type of repeated querying is typically necessary in black-box settings to characterize the response distribution $ F_{ij} $. We recognize that all of this notation can be quite cumbersome, but we believe it to be necessary when first describing the setting and methodology.
>
> > The second paragraph ...
>
> After introducing the necessary notation, we wanted to re-orient the reader to our goal and to let them know that we sometimes are loose with what variables represent (either a realization of the random variable X or the random variable X) before proceeding.
>
> > Line 102 ...
>
> Yes, it is quite a mouthful -- though we tried to make it precise. Can you suggest an alternative?
>
> > Lines 189-192 ...
>
> We wanted to be careful when presenting the theoretical results because consistency results are oftentimes confused with optimality results. In our case, we included the quotation marks because there are different notions of optimality here -- optimality with respect to the class of decision functions, optimality with respect to the distribution $ P_{\psi Y} $, and optimality with respect to the distribution $ P_{f Y} $.
>
> We plan to improve our discussion of these nuances to allow for discussion of optimality without quotation marks.
>
> > In general, from the text after theorems 1 and 2 ...
>
> That is a fair point. Again, we wanted to be careful when presenting the results so as to not oversell them. The results are actually quite informative and impactful -- Theorem 1 says that inference in estimated DKPS is consistent for inference in the true-but-unknown DKPS. Theorem 2 says if a sequence of decision functions (such as k-NN, SVMs, etc.) is known to be consistent for $ P_{\psi y} $ when trained on the true-but-unknown $ \psi $, then the analogous sequence of classifiers constructed with the estimated DKPS is also consistent for $ P_{\psi y} $. Essentially, Theorem 1 comments on the effect of $ m $ and Theorem 2 comments on the effect of $ n $. We see both of the theoretical results in action in the “Was RA Fisher great?” and “Has a model had access to sensitive-data data” examples: as m gets bigger the performance improves (Theorem 1) and as n gets bigger the performance improves (Theorem 2).
>
> We plan to improve our discussion around the Theorem results, as well, to include more insights as to what they *do* say, as well as keeping the current discussion around what they *do not* say.
>
> > Line 796 ...
>
> Our goal with appending fruit names to the end of the original 50 augmentations was to perturb the original augmentations slightly so as to get relatively densely populated regions of the DKPS. It is a cheap way to get more “models” for which the behavior in DKPS would be easy to interpret. We thought that this was particularly important given that the treatment of our setting as a classical supervised learning problem is relatively new.
>
>
> > Overall ...
>
> Thank you for your helpful feedback! Please let us know if our responses were helpful in understanding our work.
>
> $ \textbf{Questions} $:
>
> Please let us know if you have any other questions throughout this discussion period!

---

> > ### Comment · Reviewer_9WTf · 2024-12-02
> >
> > Dear authors, I thank you for your clarifications! I believe that your work has potential, but it seems to me that in its current version it is still not ready for publication, so I, unfortunately, have to leave the score unchanged.

---

### Official Review · Reviewer_SsgE · 2024-11-05

**Soundness:** 4
**Presentation:** 3
**Contribution:** 4
**Rating:** 8
**Confidence:** 2

**Summary:**

Models have a set of attributes, such as the score on a benchmark or an indicator of whether they were fit to sensitive data. These attributes may or may not be known by the user. Sometimes similar models may be used to predict such model level attributes. But how is similarity defined? The authors propose using the data kernel perspective space as a basis for similarity. They demonstrate that similarity in this space is predictive of model level attributes.

**Strengths:**

Practical as it only needs generated responses and an embedding function.
Theoretically grounded.
Good empirical evaluation using relevant examples.

**Weaknesses:**

More discussion and evaluation on the choice of embedding function would strengthen the paper. It is not yet evident what makes a good or bad embedding function. For example, is it sufficient to use a random LLM? Could one use off-the-shelf token embeddings? For each experiment, an analysis of result sensitivity to this choice would be of practical interest.

The abstract is currently a bit of a disservice to the paper. For example, while it may be evident to those familiar with the data kernel perspective that access to a target or training model's internals is not needed, when you use terms like "embedding based representations" it gives the impression that such access is needed. I think simplifying the language, emphasizing the problem you are trying to solve (model attribute inference), then illustrating why it is important, and summarizing your approach and its positive features would help.

**Questions:**

I have no further questions at this point.

---

> ### Author Response · Authors · 2024-11-14
>
> SsgE,
>
> Thank you for taking the time to read and review our paper. We have addressed your review below.
>
> $ \textbf{Summary} $:
>
> We think your summary is a fair description of our work.
>
> $ \textbf{Strengths} $:
>
> Thank you -- we also think that the method is practical & theoretically grounded and we are excited to continue to push this work in various directions!
>
> $ \textbf{Weaknesses} $:
>
> We plan to add more discussion on the effects of different hyperparameter choices (such as the embedding function, the metric on the representations of the models, etc.). In experiments not included in the current paper, we have seen that performance can be highly affected by the choice of embedding function. For example, using off-the-shelf token embeddings -- as opposed to purpose-built (though also off-the-shelf) embedding functions such as sentence-transformers/all-MiniLM-L6-v2 or nomic-ai/nomic-embed-text-v1.5 -- can cause a sharp performance drop. The process to go from collections of response embeddings for each model to a collection model embeddings is quite rich in the type and number of impactful hyperparameters.
>
> & thank you for suggesting an abstract re-write to better reach our audience.
>
> $ \textbf{Questions} $:
>
> Please let us know if any questions come up throughout the discussion period.

---

> > ### Comment · Reviewer_SsgE · 2024-11-26
> >
> > Thank you for responding.
> >
> > I am happy to maintain my score. However, there was an opportunity to update the paper during the discussion period, which doesn't appear to have been taken yet to directly address the concerns raised by myself and the other reviewers. I think the other reviewers have made reasonable requests and I would like to see some concrete updates to the manuscript to address these.

---

> ### Author Response · Authors · 2024-11-27
> **updated paper**
>
> We have just updated the paper.
> We spent the majority of our revision time re-writing the abstract and updating Section 2:
>
> $ \textbf{1. Updates to the abstract} $
>
> - Per your suggestions, we dropped references of "embedding-based representations of the models" in favor of "vector representations of black-box model". We think that this change helps emphasize that the method we does not rely on access to the model weights, token probabilities, etc. and is broadly applicable.
>
> $ \textbf{2. Updates to Section 2} $
>
> - We eliminated unnecessary notation and updated the description of the setting to improve readability.
> - We updated the presentation of classical statistical inference on generative models to flow better.
> - We added paragraphs after both theorems to highlight how they can be interpreted in practice.
> - We moved the discussion of what the theorems *do not* say to a new subsection called "Non-standard considerations". We think that this will help the reader understand the nuances of our setting.
>
> $ \textbf{3. Misc.} $
>
> - We added a "Contribution" paragraph at the end of the introduction. We hope that this helps readers understand our contribution and how we view the work in the context of recent literature. This is the full paragraph:
> > Our contribution is the theoretical and empirical validation for using vector representations of black-box generative models for model-level inference tasks. The representations that we study herein are based on the embeddings of their responses. The theoretical results apply to any generic, well-behaved embedding function. Given the representations, any standard vector-based method can be used for inference on the models.
> We think that this addresses some of your comments that one could use other inference methods for some of the tasks we study.
>
> - We clarified that the model bias and toxicity regression results reported are the average over all models in the model graph.
> - We fixed various typos.

---

### Note · Authors · 2024-12-04

**Comment:**

Thank you all for taking the time to review our paper.
We plan to incorporate your feedback in future versions of this work -- in particular, we will add more comprehensive investigations into the effects of different hyperparameters and include any relevant comparative methods.

**Withdrawal Confirmation:**

I have read and agree with the venue's withdrawal policy on behalf of myself and my co-authors.